# Electrodermal Activity during Blood Pooling for Arterial Blood Gases Analysis in Sedated Adult Intensive Care Unit Patients

**DOI:** 10.3390/medsci6010020

**Published:** 2018-03-06

**Authors:** Theodoros Aslanidis, Vasilios Grosomanidis, Konstantinos Karakoulas, Athanasios Chatzisotiriou

**Affiliations:** 1Intensive Care Unit, Department of Anesthesiology and Intensive Care Medicine, AHEPA General University Hospital, St.Kiriakidis 1, P.C 54636 Thessaloniki, Greece; 2Cardiothoracic Anesthesia Unit, Department of Anesthesiology and Intensive Care Medicine, AHEPA General University Hospital, St.Kiriakidis 1, P.C. 54636 Thessaloniki, Greece; grosoman@otenet.gr; 3Department of Anesthesiology and Intensive Care Medicine, AHEPA General University Hospital, St.Kiriakidis 1, P.C. 54636 Thessaloniki, Greece; carakoul@auth.gr; 4Laboratory of Physiology, Medical School, Aristotle University of Thessaloniki, St.Kiriakidis 1, P.C. 54636 Thessaloniki, Greece; achatzisot@med.auth.gr

**Keywords:** electrodermal activity, pain, intensive care, stress

## Abstract

Electrodermal activity (EDA) is considered a measure of autonomous nervous system activity. This study performed an exploratory analysis of the EDA changes during blood pooling for arterial blood gas analysis in sedated adult critical care patients and correlated the variations to other monitored parameters. EDA, along with other parameters, were monitored during 4 h routine daytime intensive care nursing and treatment in an adult ICU. 4 h measurements were divided into two groups based upon the sedation level. Selected recordings before and after blood pooling for arterial blood gases analysis (stress event) was performed. Nine stress events from Group A and 17 from Group B were included for further analysis. Patients’ demographics, laboratory exams, and severity scores were recorded. For both sedation levels, EDA changes are much greater than any other monitoring parameters used. The changes are noticed in both measurement (15 s and 60 s), even though in the 60 s measurement only selected EDA parameters are significantly changed after the start of the procedure. EDA measurements are more sensitive to a given stress event than cardiovascular or respiratory parameters. However, the present results could only be considered as a pilot study. More studies are needed in order to identify the real stress-load and clinical significance of such stimuli, which are considered otherwise painless in those patients.

## 1. Introduction

Arterial blood gas measurements (ABGs) are ordered as part of the regular clinical assessment of critically ill patients. Thus, they are the most common tests performed in an intensive care unit (ICU). Although the procedure is unpleasant and stressful for the wards’ patients, the existence of an arterial line in situ in most ICU patients facilitates the exam, as it is assumed to be painless. That may explain the lack of literature about the assessment of the procedure as a stress-load event.

Electrodermal activity (EDA) is a general term, which includes all electrical properties which can be traced back to the skin and its appendages. The major function of sweating is the regulation of the body temperature. Yet, it is known that sweating on the palm may be elicited by emotional, physiological, and stressful stimuli [1]. Central innervations of sweat gland activity point to several centers, located at different levels of the central nervous system (CNS), mainly the sympathetic nervous system (SNS). Hence, the activity from SNS regulates the secretory part of the sweat glands, which in turn changes the electrical properties of the skin due to the filling of electrolyte-containing sweat in the ducts. Measurement of the output of the sweat glands, which EDA is thought to do, provides a simple gauge of the level and extent of sympathetic activity. This is the simple and basic concept underlying EDA and its applications [2].

In the present pilot study, we monitor EDA changes during blood pooling for arterial blood gases analysis from an artery line in situ in sedated adult ICU patients. The phenomenon is well studied in the field of neuropsychology and psychiatry, yet reports for its perioperative applications are limited [3]. Until the present study (2013), there was only one published report about application of EDA in adult ICU patients. Yet, it focused on one parameter of available EDA variables for recording (skin conductance variability) [4]. The rest of the available data comes either for different settings (operative room or postoperative unit) or from pediatric patients. Moreover, their design aimed at measuring EDA during a variety of stimuli [5,6,7,8,9]. Thus, their conclusion could not be considered as applicable to sedated adult ICU patients.

## 2. Materials and Methods

This prospective observational study was conducted at the adult general ICU, at American Hellenic Educational Progressive Association (AHEPA) General University Hospital, Thessaloniki, Greece. The study is part of a thesis project, approved by AHEPA General University Hospital Research Committee and by no. 16/09-07-2013 General Assembly of Special Composition of Medical School, Aristotle University of Thessaloniki (ref. no. 8220/10-07-2013). A total 25 measurements in 25 critically ill patients under sedation above 18 years old were included in the study. Other inclusion criteria included administered mechanical ventilation > 24 h and constant sedation level under midazolam or propofol continuous intravenous infusion (c.i.v.). On the contrary, patients with Ramsay sedation score (RSS) 1, diagnosed or with history of hearing problems, psychiatric disorders, neurological diseases, neuro or myopathy, delirium, CNS, or spinal cord injury were excluded. Also as exclusion criteria were considered pregnancy, hemodynamic/respiratory instability, edema of the upper limbs (place of measurement) and the presence of sensitive electrical life-sustainable devices such as pacemakers, renal replacement therapy devices, intra-abdominal aortal counterpulsion pumps, extracorporal membrane oxygenators, and artificial livers.

Though a bispectral index monitor device was available, clinical priority was given to other patients; thus, it was not used in the measurements. Other methods that are suggested by the literature as autonomic nervous monitors (digital pupillometry, entropy, heart rate variability, electroencephalography EEG) were not available during the conduct of the study.

Skin conductance (SC) parameters, and selected hemodynamic and respiratory parameters (HR—heart rate, VPC—ventricular premature contractions (number), STII—electrocardiographic ST wave deviation in II lead, SAP—systolic arterial pressure, MAP—mean arterial pressure, DAP—diastolic arterial pressure, RR—respiratory rate) were monitored during 4 h routine daytime intensive care nursing and treatment (afternoon shift, measurements during 4:00 p.m.–8:00 p.m.). Measurements were divided into two categories according to patients’ sedation level: Group ARSS 2–4 (n_a_ = 10) and Group B—RSS 5–6 (n_b_ = 15). Dosing to achieve the given sedation level, although recorded, was not taken into account (since a point of interest was sedation level).

A Med Storm Pain Monitor System (MED Storm Innovation AS, Oslo, Norway) was used as the SC monitor [10]. Three single-use Ag/Cl electrodes were attached at the palmar surface of the hand: on the thenar eminence (stimulator current), on the hypothenar eminence (recording electrode), and just below second and third digits (reference). In order to minimize artifacts, the hand least likely to move, with no intravenous or intra-arterial lines was chosen. SC was measured by alternating current of 66 Hz (V) and an applied voltage of 50 mV. SC parameters recorded were: absolute SC (in μS), peaks/sec or number of SC fluctuations per second (NSCF), the average peak (AvP) **(**micro Siemens seconds—μSs), the rate of increase or decrease from the start to the end of the measurement window (rise time, AvRT, in micro Siemens per second—μS/s), area huge peaks (ArHP) (μSs), area small peaks (ArSP) (μSs), and the larger of the two measures (referred to as the area under curve—AUC, in μSs). Cut off for NSCF counting was >0.005, much more sensitive than the >0.02 μS used in relative pain monitoring literature [11,12]. Signal quality <80% was considered artifact and the measurement was also excluded [10] (see Appendix A’ images).

Two measurement windows of interest were used: (1) 15 s before and 15 s (pre-set window by the given monitor for measuring effect of short lasting stimuli) after and (2) 60 s before and 60 s after blood pooling from arterial line in situ for arterial blood gases exam. The 60 s window was chosen in order to average out the effect. Recording was considered for further analysis if only 4 min before and 1 min after the stimulus there was no other stimulus of any kind (i.e., alarm noise). In order to ensure the observational character of the study, the stimulation event (referred to as the ‘ABG event’) was a product of the daily nursing/treatment routing inside ICU environment and not artificial deliberately-created stimulus.

The rest of the parameters were monitored via Bedside Monitor BSM 9101K and Monitor CNS 9601 (Nihon Kohden Ltd., Tokyo, Japan); and included: heart rate (HR), systolic (SAP), diastolic (DAP) and mean arterial pressure (MAP), number of ventricular premature contractions (VPC), electocardiographic ST wave deviation in II lead (ST II) and respiratory rate (RR). Since the above were used in the literature [3] as possible measures of stress, recordings were used as measure of comparison with SC parameters. Ambient noise level was measured at distance 30 cm from the head of the patient via Sound Level Meter GM13656 (Shenzhen Jumaoyuan Science & Technology Co., Shenzhen, China) (Figure 1).

Data analysis was performed with MS Office Excel 2007 (Microsoft Co., Washington, DC, USA) and Rstudio IDE v.1.00.136 for R v.3.3.2 (Rstudio Inc., Boston, FL, USA).

Descriptive statistics are presented as weighted average ($\bar{x}$), standard deviation (*s*), and first and third quartile (*Q*_1_ and *Q*_3_ respectively). Two comparison designs were applied: one examined acute changes before/after the noise stimulus and one that examined the range of change between the two groups. Shapiro–Francia normality test is performed for the parameters of interest and then paired Student’s *t*-test or Wilcoxon signed ranked test is calculated. Results are presented as *p* value (confidence interval—CI). Statistical significance for *p* is set to *p* < 0.05 and CI level at 95%.

## 3. Results

General characteristic of patients in each group of measurements is illustrated in Table 1. Different averages of APACHE II (Acute Physiology and Chronic Health disease Classification System II) score, Extended Glasgow Outcome Score (GOSE) and PaO_2_/FiO_2_ are partially explain the different sedation level. All measurements were conducted on white Caucasian patients. Ambient noise levels, 4 min before the start of the procedure were: 57.53 (4.75) dB in Group A and 56.54 (2.62) dB in Group B. Hemoglobin and serum electrolytes were within normal limits for both groups.

During recording time, 9 ‘ABG events’ took place in Group A and 17 in Group B that met inclusion criteria for further analysis.

In Group A, all the EDA parameters (ArHP, ArSP, NFSC, AvRT, AvP, AUC, and SC) were significantly changed (in all parameters *p* < 0.03, see Appendix A*)* at the 15 s measurements, while at the same time the rest of the measured variables stayed relatively constant. The changes persisted over time: thus, most variables’ measurements (ArHP, NFSC, AvP, AUC) also displayed significant (almost in the same extend) change at the 60 s measurement.

Despite the deepest sedation level, similar findings were recorded in Group B with the selected EDA parameters. In the 15 s measurements, NFSC, AvP, AUC, and SC were significantly changed (*p* < 0.03, see Appendix A), while ArHP, NFSC, AUC, and SC are the variables with the greatest change.

## 4. Discussion

The above results display some interesting findings. For both levels of sedation, EDA changes are much greater than any other monitoring parameters used. The changes are noticed in both measurement (15 s and 60 s), even though in the 60 s measurement, only selected EDA parameters are significantly changed after the start of the procedure. The authors assume that, since the patients in Group A were in the position of knowing that the exam is about to take place, the stimulus itself had lower stress ‘load’, and this was reflected in the number of EDA parameters that had changed.

Apart from that, though the criteria for including the measurements of a procedure were relatively strict, performing the exam to more awake patients may affect the procedure itself in ways other than those predefined by the authors (pressure applied to blood pooling, etc.).

All measurements were conducted on white Caucasian patient and both groups were similar in age, weight, and body mass index (BMI). Thus, biases from the aforementioned factors were absent [12]. The same is also valid for ambient noise. The latter is considered also important, since noise is recognized as stress stimulus in ICU [13]. Other environmental disturbances (e.g., ambient temperature) were considered minimal as the study took place in an ICU (= controlled) environment.

Sex may play a confounding role in EDA measurement because of monthly hormonal variations in women [14]; yet, laboratory studies in ambulatory setting have been inconclusive [12,15]. The measurements in the present study were conducted in older women.

Sleep quality has been connected in the literature with several diseases [16]. Bad sleep quality may affect Autonomic Nervous System (ANS) activity, and consequently, EDA measurements. Thus, quality of sleep between the two groups is a possible contributing factor; however, it was not evaluated in the current study.

Other limitations are the relatively small number of measurements and the open, observational character of the study. The results do reveal a certain stress-load of the procedure in interest; yet further conclusions are unsafe. Is there any pain from a procedure that is considered painless in sedated patients with an arterial line in situ? No matter the existence of literature about the use of EDA as pain index, for the time being (2017) there are no conclusive reports about relations of EDA recordings with certain pain levels [3]. More generally, we cannot determine the negative stress-load of these findings. The exact role and physiological ‘reflection’ of each of the aforementioned EDA parameters to the ANS activity is yet to be determined [16,17,18].

Further studies in bigger samples both in ‘stable’ and ‘unstable’ ICU patients, probably with more predefined stimuli, are needed in order to have a clear idea of the role of EDA monitoring in adult ICU environment. Still, there is little data about ICU patients. In the latter field, it has been mainly used as analgesia monitor; but results are inconclusive. Recently, one trial studied EDA monitoring during tube removal [19], while another one suggested its use as biomarker of outcome after cardiac arrest [20]. Results from the same ICU examining more stimuli, such as suction or noise alarms, are expected to be published. Longer measurements intervals are also necessary in order to examine the habituation (if any) of ANS in a specific stimulus and to evaluate any possible repeatable patterns. However, this is extremely difficult in the rapidly dynamic setting of an intensive care unit. Furthermore, technical requirements for the device used for a 24 h measurement, for example, may be different from those applied in the present study.

## 5. Conclusions

Electrodermal activity measurements are more sensitive to the procedure of blood pooling for ABG analysis in sedated adult ICU patients than cardiovascular and respiratory monitoring, thus serving as a more sensitive index of stimulus-induced stress. However, future studies are needed in order to define EDA role as stress monitor and to clarify possible specific stimulus EDA response patterns in all group of ICU patients.

## Figures and Tables

**Figure 1 medsci-06-00020-f001:**
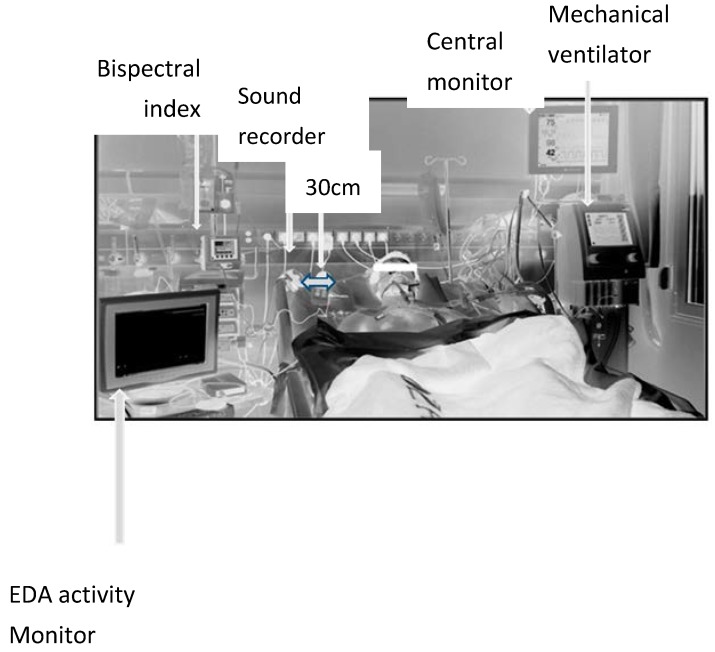
Arrangement of recordings devices during the observation period.

**Table 1 medsci-06-00020-t001:** General characteristics of the patients included finally in each group

	Group A	Group B		Group A	Group B
**(*n)* measure**	10	15	**APACHE II**	15.4 (1.55)	19.6 (1.66)
**Sex**	Male = 10, Female = 0	Male = 9, Female = 6	**SOFA**	6.3 (0.9)	7.9 (0.4)
**Age (years)**	66.5 (14.8)	63.8 (10.9)	**GOSE**	6.4 (0.9)	5.2 (0.8)
**Weight (kg)**	90.6 (15.1)	89.95 (12.6)	**t (°C)**	37.2 (0.3)	37.1 (0.4)
**ΒMI (kg/m^2^)**	28 (1.65)	30.3 (0.85)	**PaO_2_/FiO_2_**	294 (69.3)	230 (81.8)

Presented form: mean (SD), rounded to the nearest decimal. SOFA: Sequential Organ Failure Assessment (SOFA) Score. APACHE II: Acute Physiology and Chronic Health disease Classification System II, GOSE: Extended Glasgow Outcome Score, BMI: body mass index, t: temperature.

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
