# Peer review of "Electrodermal Activity during Blood Pooling for Arterial Blood Gases Analysis in Sedated Adult Intensive Care Unit Patients"

_medsci, 2018, doi:10.3390/medsci6010020_

Round 1
Reviewer 1 Report
"This paper is of importance due to the validation of skin conductance to assess pain in intensive care unit patients. It is of high importance to tailor the level of analgesia in intensive care units patients to avoid side effects. Also the opioid-epidemic has given this topic high importance. The skin conductance variables will assess pain in patients who are circulatory instable, like ECMO-treatment or patients who use epinephrine or beta-blocker as well as alfa-2 agonists due to the fact that skin conductance mirror skin sympathetic nerves where acetyl choline acts on muscarinic receptors, and is therefore not influenced from changes in blood circulation. Since the skin conductance method acts fast, within 1-2 sec, and it is low variation between individuals when they are at the same pain level, it can be used as an index valid for different patients groups also including intensive care unit patients."
NSCF should be peaks/sec not micorsiemenssec
2. Comment on that the 15 sec analysing window which is a pre-set window to assess also short lasting painful stimuli. The 60 sec window is to assess longer lasting painful stimuli (short lasting painful stimuli will be averaged out in a 60 sec analysing window).
2. Include more references on previously validation of the NSCF: 1. Gjerstad AC, Wagner K, Henrichsen T, Storm H. Skin Conductance Versus the Modified COMFORT Sedation Score as a Measured of Discomfort in Artificially Ventilated Children. Abstract ESA 2002, Pediatrics 2008;122;e848-e853.
2. Gunther A, Sackey P, Karolinsk hospital, Storm H. Skin conductance changes in intensive care patients. Critical Care 2013, 17:R51.
3. www.kib.ki.se “Skin conductance variability and stressful exposures in critical care” (Thesis from Anders Gunther MD.PhD)
4. Gunther A. Hansen JO, Sackey P, Storm H, Bernhardsson J, Sundin Ø, Bjärtå A. Measuring pain – a validation of physiological and self-rated measurements, and an investigation of the relationship between them. ESICM Lives 2016 i October: https://services.y-congress.com/congress/y-upload/index.aspx?sid=681UQNUNc7A%3d&client=esicm
5. Thomas Ledowski, Hanne Storm et al. Monitoring of skin conductance to assess postoperative pain intensity. British Journal of Anaesthesia 2006;97(6):862-5.
6. Ledowski T, Bromilow J, Wu J, Paech MJ, Storm H, Schug SA. The assessment of postoperative pain by monitoring skin conductance: results of a prospective study, Anaesthesia. 2007 Oct;62(10):989-93
7. Hullett B, Chambers N, Preuss J, Zamudio I, Lange J, Pascoe E, Ledowski T, Monitoring Electrical Skin Conductance A Tool for the Assessment of Postoperative Pain in Children? Anesthesiology 2009; 111:513–7
Author Response
Dear reviewer,
thank you very much for the comments you made.
I have corrected the unit in NSCF, commented the selected time windows and included more than the proposed references in the text.
Please, see the revised manuscript at the file uploaded.
With kindest regards,
Theodoros Aslanidis

Reviewer 2 Report
This paper presents a performed study about an exploratory analysis of the EDA changes during 21 blood pooling for arterial blood gas analysis in sedated adult critical care patients and correlated 22 the variations to other monitored parameters.
The scientific idea of the present research is very interesting, but I think the paper has too many gaps.
In the introduction section, an adequate description of the reference literature is missing. There are no studies related to the research, there are only two references. So I do not understand the novelty reported by this study. Furthermore, the authors write “The major function of sweating is the regulation of the body temperature. Yet, it is known that sweating on the palm may be independent of the ambient temperature [… ].” This assumption on what is based? on which bibliographic reference? In fact it is known that it is influenced by the temperature of the environment, which is why conductivity studies must always be performed in a room with controlled environmental temperature. Did the authors do this?
In the Materials and Methods section, the authors write “Two measurement window of interest were used: 1st) 15sec before and 15sec after and 2nd 60 sec before and 60sec after blood pooling from arterial line in situ for Arterial blood gases exam. Recording was considered for further analysis if only 4min before and 1 min after the stimulus there was no other stimulus of any kind (i.e. alarm noise) [..].Why the authors have chosen these times? On which bibliographical reference?
The indications on the procedure of Skin conductance are missing, as follows: if the recordings were made in the same time slot for all the participants, in the same season etc etc. Furthermore, I know how difficult it is to do research on patients but the sample size is too low, at best it could be a pilot study.
In the Discussion section, the discussion of the results compared to the international literature is missing.
Author Response
Dear reviewer,
thank you very much for the comments you made.
The study took place in a controlled environement (ICU, ambient temperature of 23-24oC and relative humidity of around 50%).
I also have included comments both of the other notes you mentioned.
Please, see the revised manuscript at the file uploaded.
With kindest regards,
Theodoros Aslanidis

Reviewer 3 Report
Electrodermal activity during blood pooling for arterial blood gases analysis in sedated adult Intensive Care Unit patients
There are many questions I have about this paper.
Firstly, there was insufficient detail about the objective of the study. Whilst I understand that patients experience stress in ICU, what would be the objective of measuring the stress in this manner. Would the authors aim be to provide methods to alleviate the stress and would this affect outcome. Would the authors be able to identify the frequency of stressors during a longer period such as 24 hours? This stressor was measured for 60 seconds wither side of the intervention. I am not provided with the variation in SC during a longer period.
Secondly, whilst I understand about the process of arterial blood gas sampling, what do the authors mean by blood pooling. Perhaps an image or diagram would help the naïve reader.
Thirdly, if the authors wished to validate a stress monitor (Skin Conductance - SC) in Intensive Care, why did they choose a procedure that is not likely to invoke stress in the individual? Would suctioning or IV access, or chest physiotherapy or other uncomfortable procedures be more likely to confirm the value of SC? What is the change in SC over time, and is the change during blood sampling a small change or a large change. How would it compare to line insertion, chest physiotherapy or other noxious interventions.
Looking at the other studies using SC for monitoring conscious state, is SC measuring arousal or an unpleasant sensation. Can it differentiate between an increase state of awareness or anxiety and distress? There appears to be an assumption implicit in the study that the patient is experiencing distress. How could the authors clarify this difference?
The Ramsay sedation score is not discriminatory when looking in detail at levels of sedation and arousal. The state of consciousness is a continuous variable that is dependent on many factors. The RSS is a discrete variable, which I think is not suited to this study. Could the authors not use BIS or other EEG variables to use as a measure of conscious state, as other authors looking at SC in the peri-operative setting?
If the objective of the study was to demonstrate that blood pooling was a stressful procedure, why demonstrate the difference between two groups. Should the initial data examination explore if in the less sedated group the SC is demonstrably different before and after the procedure. I am unsure what the second more sedated group contributed to the study. Could the research group have chosen a group of patients that were not sedated on intensive care and assess the response to blood pooling?
Do Propofol and Midazolam influence the SC variability in a similar manner? What effect do other drugs have on SC during intensive care? I would include the sympathomimetics, beta blockers, etc. The authors have not defined the amount of midazolam and propofol or the duration of time the infusion was maintained. There is no qualification of the reasons for the patient admission to intensive care.
Regarding the signal quality, would the authors report the frequency that the signal quality fell below 80%. How long was the measurement time? The authors state that the measurement was taken 15 seconds before and after and 60 seconds before and after an intervention. Was the measurement time averaged over several seconds? Other authors have reported signal measurement time of up to several minutes. Does the very short signal measurement time influence the results?
The table of results has the heart rate and blood pressure average to one or two decimal places. This is not logical as these values are not recorded at one or two decimal places.
The table legend describes a BIS value. I cannot see this value in the table and I do not recall this value being described in the methods section.
Regarding SC, the authors did not qualify in the methods what would be considered a significant change in SC when considering a painful intervention. The NFSC appears to be no more than 0.15. The study by Gunther suggested this was only a mild stimulus and not painful[1]. Though I may misunderstand the data presented.
Overall, whilst I appreciate the effort that the authors have put into the study, there are elements of the design that are not well considered. The authors have not made clear the objectives of measuring the blood sampling procedure in sedated patients, over other more painful procedures. The authors have used other physiological variables to compare with SC, but it is known that these variables are not reliable at detecting a change in emotional state. The results and methods do not describe measures of EEG activity, such as BIS or Entropy, which may be another practical surrogate marker of procedural stress.
[1] Crit Care. 2013 Mar 19;17(2):R51 Palmar skin conductance variability and the relation to stimulation, pain and the motor activity assessment scale in intensive care unit patients. Günther AC, Bottai M, Schandl AR, Storm H, Rossi P, Sackey PV.
Author Response
Dear reviewer,
Thank you very much for the comments on the manuscript.
I have modified the text in order to clarify the points you mentioned.
Yet, allow me to highlight them in brief:
-Considering the use for longer terms (24h ) , there has never been application of the given device for long term monitor. Other technical requirements may needed; there are available devices in the market, used mainly for epileptics (but with no safe global database about) .However , there was not such a device available.
- Not availability reason and character of the study (observational) explains the luck of use of monitors like entropy, EEG, HRV, PRV and digital pupillometry. BIS was available, but a) clinical priority had been given over research priority b)several limitation are need to be considered in its use in ICU.
- 15sec interval is a preset window for short acting stimuli, 60sec was set in order to “catch” the peaks of the effect. We are planning a future study for studying habituation of ANS and possible patterns. The same was considered for the 80% cutoff signal limit value.
- The text is part of a larger project. Data gathered exceed the allowed length of an article. More stimulis- as suggested in the review- are already studied; yet publication of them is pending. The stimulus selected in the presented article is interesting for that exactly the reason- we do not consider it strong stimulus in ICU patients, sedated and with an arterial line in situ. The fact that such “innocent” stimuli elicit ANS reaction, “caught” in EDA measurements, is something that may open a new perspective.
- The relative small sample is due to the observational character of the study and the strict inclusion criteria. We are more than sure that a different study design is needed; yet RCT in ICU patients is something easy said, extremely difficult to conduct. The data presented considered the “ABG Events” that fitted with the inclusion criteria. En plus, since in first place there was no enough data about ICU patients; we considered it logical to conduct first an observational study in real conditions and then design a RCT.
- Till the conduct of the study (2013) , data were scarce about adult ICU patients. Thus the data presented by Gunther et al (Crit Care 2013), could not serve as reliable and enough in quantity data to be based in. Data from the rest of perioperative setting, present a variety of designs, populations and measurements.
Please see the revised manuscript.
With Kindest Regards,
Theodoros Aslanidis

Round 2
Reviewer 2 Report
I appreciated the changes made by the authors. The manuscript has improved a lot in its quality. In my opinion it is accepted in this form.